# Comparison of the Use of Desflurane vs. Propofol in Aortic Valve Replacement Surgery: Differences in Nephroprotection: An Explorative and Hypothesis-Generating Study

**DOI:** 10.3390/life12081172

**Published:** 2022-07-31

**Authors:** Jose Luis Guerrero Orriach, Alfredo Malo-Manso, Mercedes Nuñez Galo, Inmaculada Bellido Estevez, Amalio Ruiz Salas, Jose Cruz Mañas, Lourdes Garrido-Sanchez, Laura Gonzalez-Alvarez

**Affiliations:** 1Instituto de Investigación Biomédica de Málaga y Plataforma en Nanomedicina-IBIMA Plataforma BIONAND, 29010 Malaga, Spain; alfredomalomanso@gmail.com (A.M.-M.); ibellido@uma.es (I.B.E.); amalio_ruiz@hotmail.com (A.R.S.); jose.cruz.sspa@juntadeandalucia.es (J.C.M.); lourgarrido@hotmail.com (L.G.-S.); 2Department of Anesthesiology, Virgen de la Victoria University Hospital, 29010 Malaga, Spain; mng_rem@hotmail.com; 3Department of Pharmacology and Pediatrics, School of Medicine, University of Malaga, 29010 Malaga, Spain; 4Unidad de Gestión Clínica de Endocrinología y Nutrición, Hospital Universitario Virgen de la Victoria, 29010 Malaga, Spain; 5CIBER Fisiopatología de la Obesidad y Nutrición (CIBERObn), Instituto Salud Carlos III, 28029 Madrid, Spain

**Keywords:** desflurane, cardiac surgery, halogenated, kidney, heart, preconditioning

## Abstract

*Introduction*: The cardioprotective effect of halogenated drugs in cardiac surgery has been the subject of several studies. However, there is scarcity of data on their potential nephroprotective effects. Aortic valve replacement and coronary revascularization are the most frequent cardiac surgery procedures. The objective of this explorative study was to examine the effect of desflurane vs. propofol on renal function, when administered in aortic valve replacement surgery, including the extracorporeal circulation period. *Method*: A quasi-experimental prospective study was performed in 60 patients, who were allocated to receive either desflurane or propofol intraoperatively during aortic valve replacement surgery. As a hypnotic, group 1 received propofol, whereas group 2 received desflurane. Markers of renal function and level of cardiac preservation were determined based on biochemical parameters (troponin I, NTProBNP). *Results*: In the propofol group, there were significant variations between postoperative values of urinary NGAL and creatinine and baseline values. In contrast, no variations were found in the desflurane group in terms of hemodynamic parameters and myocardial damage. *Conclusions*: The use of propofol vs. desflurane during aortic valve replacement surgery is associated with a decrease in renal function.

## 1. Introduction

The intraoperative use of halogenated agents in cardiac surgery provides multiple benefits, primarily on cardiac function [1,2,3]. Cardioprotection is exerted by desflurane through pre- and postconditioning, during the perioperative period of cardiac surgery involving induced ischemia (during extracorporeal circulation (ECC)) or when there is a high risk of coronary events secondary to ischemic heart disease [4]. ECC increases the risk of myocardial damage caused by existing heart disease or by a potential imbalance between oxygen supply and consumption [2,5,6,7,8]. As opposed to intravenous agents (propofol), inhaled anesthetics amplify enzymatic pathways according to the time of exposure and concentration administered; therefore, sustained perioperative administration should enhance the protective effects of inhaled anesthetics.

The use of halogenated agents in the perioperative period of cardiac surgery is associated with a reduction of low cardiac output syndrome (LCOS) and lower levels of myocardial damage markers [9,10].

The perioperative role of halogenated agents in organ protection has been the subject of study in the recent years. Thus, numerous studies have focused on their effects on the kidneys, due to their relationship with mortality during the postoperative period of cardiac surgery. Hence, different investigations correlate the use of halogenated agents with improved renal function due to their two-fold role. Firstly, they induce myocardial conditioning, thereby improving cardiac output and ultimately preserving cardiac function. Secondly, they provide nephroprotection through a similar mechanism [6,10,11].

Of the range of drugs used in cardiac surgery, studies most frequently focus on the beneficial effects of sevoflurane. However, the role of this agent during cardiopulmonary bypass has only been evaluated in a few studies, of which primary objective was to assess its cardioprotective effects [12].

Our primary objective was to examine the association between variations in kidney function in patients who received propofol vs. desflurane for anesthesia. Thus, variations were determined on the basis of associated biochemical and clinical parameters (i.e., creatinine, diuresis and NGAL). Our secondary objective was to compare the level of cardiac preservation in the two study groups to understand the biochemical (troponin I, NTProBNP) and clinical impact of the intraoperative use of desflurane, as compared to propofol.

## 2. Materials and Methods

A prospective observational study was performed in Virgen de la Victoria hospital in Malaga, Spain. The study was approved by the *Malaga Norte* Ethics Committee (EPA SP-DESF 1/14). Informed consent was obtained from all the patients included in the study. The study was carried out in accordance with the tenets of the Declaration of Helsinki. The recruited patients were candidates for aortic valve replacement surgery and met the inclusion/exclusion criteria, as follows (Figure 1):

Inclusion criteria: 1. Elective aortic valve replacement surgery (patients undergoing a concomitant procedure were excluded). 2. EUROSCORE (European Scale of Risk, useful in the perioperative period of cardiac surgery, validated at medical and scientific level) of less than 8 (moderate cardiological risk in the perioperative period). 3. Degree of anesthetic risk according to the American Society of Anesthesia (ASA) below or equal to 4 (patient with moderate-high anesthetic risk).

Exclusion criteria: 1. Clinical history of adverse reaction to different anesthetic drugs. 2. A severely diseased organ (lung, liver, kidney) diagnosed preoperatively, determined through a review of the clinical judgments recorded on the medical history of the patient. 3. Combined surgery (e.g., aortic valve and carotid surgery). 4. Patients with hemodynamic instability, heart failure or needing inotropic or vasoactive drugs prior to the intervention. 5. Treatment with oral antidiabetics not suspended at least 48 h before the procedure. 6. Treatment with euphylline/theophylline prior to the intervention.

The drug administered was determined by the anesthesiologist in charge before the beginning of the study.

Group 1: Intraoperative propofol.

Group 2: Intraoperative desflurane.

The hemodynamic management of LCOS was performed according to the consensus and recommendation document of the Spanish Society of Intensive Care [13], as follows. Diuresis <1 mL/min is managed by optimizing hemodynamics according to the protocol (treatment for LCOS and MAP is started in the presence of variations <15 with respect to baseline level). If diuresis is <1 mL/kg/h one hour after the start of the treatment, 500 mL of crystalloid are administered (Plasmalyte) in 30 min; if result is not satisfactory, the procedure is repeated as long as systolic volume variation (SVV) is <15%; in the absence of response, 10 mg of furosemide are administered at 30-min intervals until a favorable response is achieved. If urine output is >1 mL/kg/h, no diuretic drug is administered.

All patients were monitored with a 5-lead electrocardiogram. In all of them, a continuous record was kept of lead II and V, of invasive (BP) through the radial artery and of cardiac output through the Mostcare^®^ (Vygon, Italy) monitor. Pulse oximetry, capnography and blood pressure (BP) monitoring were also performed through bispectral index (BIS) hypnosis (BIS XP^®^; Aspect Medical Systems, Newton, MA, USA), provided that the patient remained sedated and connected to mechanical ventilation at an adequate concentration to maintain the level of hypnosis within the sedation ranges described (60–70). In the two groups, anesthesia was induced with etomidate 0.3 mg/kg, fentanyl 4 mcg/kg and succinylcholine 0.6 mg/kg. The different hypnotic groups were adjusted to maintain BIS within the 45–60 range. Neuromuscular relaxation was induced through intravenous administration of a bolus of cisatracurium (Nimbex^®^; Glaxosmithkline, London, UK) 0.15 mg/kg, followed by a perfusion of 1–2 mcg/kg/min in the two groups. Likewise, in all patients, the opiate of choice post-intubation was intravenous remifentanyl at an infusion rate of 0.1–0.2 mcg/kg/min (Ultiva^®^; GlaxoSmithKline, Genval, Belgium).

Epidemiological data were collected, including age, sex and EUROSCORE (risk of perioperative mortality for patients undergoing cardiac surgery). Intraoperative data included: ECC time, ischemia time.

Biochemical analysis included troponin I, CK, CKMMB, creatinine, lactate, and NT-ProBNP. Determinations were performed at diagnosis, immediately before the intervention, and during the following 48 h, at 24 h intervals. 

Urine NGAL (uNGAL) was collected at baseline and at 2 h from arrival to the Resuscitation Unit.

Hemodynamic parameters: heart rate; mean arterial pressure; right ventricular end-diastolic pressure; SatVO2, diagnosis of LCOS CI < 2 L/min/m^2^ (Mostcare^®^, Vygon, Italy); or SatVO2 < 65% after hemodynamic optimization or need for inotropic drugs. These parameters were calculated at 12-h intervals after the patient entered the operating room and for the following 48 h.

Renal function parameters: dieresis, creatinine, need for furosemide or renal replacement therapy (RRT), stage according to AKI scale (Acute Kidney Injury Scale) [14], every 6 h until 48 h after admission.

The urine NGAL ELISA was performed using a commercially available assay (NGAL ELISA Kit 036; AntibodyShop, Grusbakken, Denmark) that specifically detects human NGAL (NGAL range, 0 to 1000 ng/mL).

Urinary NGAL excretion is presented as the volume of urinary NGAL in ng per ml urine, as well as in ng per mg of urine creatinine to correct for differences in NGAL due to urine dilution. All measurements were made in triplicate. Several studies show that urine neutrophil gelatinase-associated lipocalin (NGAL) measured by ELISA is an early predictive biomarker of acute kidney injury (AKI) after cardiopulmonary bypass (CPB).

Interferences of these biomarkers occur in settings of hyperbilirubinemia, hypertriglyceridemia and intense hemolysis with elevated hemoglobinuria.

### 2.1. Sample Size Justification

According to Prabhu et al. [15], uNGAL determined 2 h after surgery has been shown to be an effective early marker of kidney injury. Sample size was calculated to distinguish a difference of 150 ng/mL in urine before and after the administration of the drug. As a result, 27 patients were necessary to achieve a beta statistical power of 80% and α = 0.05. Assuming a rate of loss to follow-up of 10%, 30 patients per group were necessary (calculations performed using statistical program GraphPad Prism 9).

### 2.2. Statistical Analysis

Normal distribution was studied using the Shapiro–Wilk normality test and sample variance distribution using the Levenne test. Results with normal or non-normal distribution but with *n* = 30 per group were analyzed using the *t*-test for normal distributions and homogeneous variances. Non-normal distributions were analyzed using the Mann–Whitney–Wilcoxon test. Binary variables were analyzed with the two-proportions z-test.

## 3. Results

There were no statistical differences between groups in hemodynamic or biochemical parameters related to myocardial damage and heart insufficiency either at baseline, at 24 h or at 48 h. In relation to kidney function, no significant differences were observed at baseline levels of NGAL and creatinine between groups. NGAL was determined at baseline and at two hours after the procedure, without any significant differences between groups in terms of baseline and postoperative values. Determination of creatinine and diuresis was performed at baseline and at 24 and 48 h, without any significant differences. Intragroup analysis revealed significant differences in the propofol group, as compared to desflurane, where differences were not observed. There were no significant differences in the administration of crystalloid solutions during the first 24 h and 48 h between groups.

There were no significant differences between groups in the epidemiological and intraoperative variables collected. (Table 1).

No differences were found in the hemodynamic parameters monitored during the study (Table 2). 

There were no significant differences between groups in renal function in terms of biochemical parameters and stage of kidney function. (Table 3).

In relation to variation in kidney function, there were significant variations in the increase in creatinine and NGAL in the propofol group at baseline, at 24 h and at 48 h, compared to the desflurane group (Figure 2 and Figure 3).

## 4. Discussion

The intraoperative use of desflurane in aortic valve replacement surgery in patients without pre-existing kidney disease has been shown to exert nephroprotective effects.

The use of halogenated drugs has been consistently associated with a cardioprotective role based on the molecular effects of pharmacological pre- and postconditioning. Its final effects are modulated by different mediator enzymes involved in the SAFE and RISK pathways [8,15]. Clinically, this phenomenon translates into a reduction in the incidence of LCOS resulting from a decrease in biochemical markers of myocardial damage (troponin I, NTProBNP). These effects result in a lower use of inotropic drugs and a shorter hospital stay [2,16]. Sample size was not calculated to assess differences between groups in terms of myocardial damage and/or lower incidence of LCOS in patients treated with desflurane as a hypnotic vs. propofol. Consequently, significant differences were not observed, although there was a tendency of desflurane to reduce cardiac damage.

Several studies have assessed the effects of halogenated hypnotics on kidney function in cardiac surgery patients. Initially, it seemed that the effects of sevoflurane on the kidneys mediated by compound A could cause toxicity and kidney injury; as a result, flow rates < 2 L/min were not recommended for anesthesia induction. Subsequent studies showed that the use of sevoflurane could be beneficial in cardiac surgery patients, since it preserves renal function during the perioperative period [6,17]. The patients who received sevoflurane as a hypnotic during the immediate intraoperative and postoperative period of myocardial revascularization surgery exhibited lower levels of uNGAL at 2 hours post-operation, as compared to the propofol group.

The mechanism by which halogenated drugs exert nephroprotective effects could be multifactorial. Firstly, decreased LCOS and preserved perfusion would result in nephropreservation [1,2,3,4]. Second, elevated central venous pressure resulting from right ventricular dysfunction or retrograde pressure causing an increase in renal venous pressure would reduce renal perfusion pressure [18,19]. Third, the mechanisms by which halogenated agents exert cardioprotection (pre- and postconditioning) also protect the kidneys. Thus, enzymes such as Akt and ERK ½ or the STAT group are also mediators of nephroprotection [20].

The purpose of this observational study was to investigate whether halogenated agents exert protective effects against kidney injury in aortic valve replacement surgery. Another objective was to assess the association between the use of halogenated agents and the incidence of low cardiac output syndrome or perioperative myocardial injury. We expect that these results open an avenue for future randomized trials testing the hypothesis that halogenated agents have a protective effect against kidney injury that is independent from their action at the myocardial level.

In our study, the propofol group showed significant intra-group variations in the levels of creatinine at baseline, at 24 and 48 h, a phenomenon that was not observed in the desflurane group. In the statistical analysis, the difference between groups at baseline, 24 and 48 h was 0.05. Despite not reaching statistical significance, an important difference was observed between the two groups in the behavior of creatinine. In fact, intra-group variations reflect a different tendency in renal function in the two groups.

In our search for a more sensitive and specific marker of renal dysfunction, we determined baseline and postoperative values of urinary NGAL in the 30 patients of each group. However, five samples of each group could not be analysed, since they were out of the limit of detection of ELISA due to hemolysis occurring during ECC [21]. Even so, we found significant variations between baseline and postoperative values in the propofol group, whereas there were no variations in the desflurane group. The results of our study demonstrate a greater benefit from the use of desflurane in all markers of renal dysfunction, as compared to propofol, something that was confirmed by intra-group analysis.

Our results show that the use of desflurane compared to propofol during a low-risk perioperative intervention in cardiac surgery generates a benefit at the kidney level, which is an independent factor of mortality in this group of patients. This potential benefit may be modulated by sustained exposure to desflurane throughout the intraoperative period of aortic valve replacement surgery, including the extracorporeal circulation period. In previous studies, the cardioprotective effects of desflurane have been associated with drug concentration and duration of exposure. The desflurane concentrations used were established for clinical purposes guided by hypnosis monitoring. Therefore, results are related to the usual use of the drug in this group of patients.

The explanation of this nephroprotective effect may be related to the cardioprotective mechanisms of desflurane. However, the treatment for LCOS was similar in the two groups, without significant variations in central venous saturation; therefore, we assume that oxygen transport was adequate in terms of oxygen consumption, and there were no differences in mean arterial pressures that would justify these differences. In addition, no differences were found either between groups in values of central venous pressure or its possible correlation with retrograde pressure. 

To the best of our knowledge, this is the first study to assess variations in NGAL associated with the use of anesthetics in patients undergoing aortic valve replacement surgery. Julier evaluated kidney function in patients undergoing off-pump myocardial revascularization surgery who received anesthesia with sevoflurane vs. propofol. Differences between our study and those of Julier [10] and Guerrero-Orriach [6] are based on the type of surgery and halogenated agent compared to propofol, and the inclusion of kidney function as primary objective.

There is little evidence on the effects of the different anesthetics on renal function in cardiac surgery patients. Indeed, assessing this association was not the primary objective in any of the studies available. Of note, Julier [10] carried out a study in patients undergoing myocardial revascularization surgery to compare the effects of sevoflurane vs. propofol on the myocardium. Levels of cardiac enzymes were lower in the group that received the halogenated agent. Likewise, significant differences were observed in cystatin C elevation, in favor of the sevoflurane group. The primary objective of our study was to assess the effects of desflurane on renal function. We chose desflurane instead of sevoflurane, which has been more widely studied, since desflurane has been suggested to have the same cardioprotective effects as sevoflurane. Although high-sensitivity biochemical markers were used in the two studies, the type of surgery and drugs administered differed. Guerrero-Orriach correlated the cardioprotective effects of sevoflurane according to the duration of exposure to the agent, as compared to propofol. The cardioprotective effect of sevoflurane improved as exposure to the agent increased. As a secondary objective, the effects of sevoflurane on renal function were assessed based on NGAL values. In agreement with our study, the authors reported that renal protection increased in patients exposed to sevoflurane for longer [6]. However, unlike our study, patients underwent myocardial revascularization surgery without extracorporeal circulation.

The reasons that raised our interest in assessing the role of desflurane in renal function in the setting of aortic valve replacement surgery were two-fold: (i) there were no studies available in which the primary objective was to examine the association between the anesthetic used during cardiac surgery and renal function; and (ii) there was a scarcity of studies examining desflurane (halogenated agent), which has been suggested to have similar cardioprotective effects as sevoflurane in cardiac surgery patients. Considering that the association between the beneficial effects of inhalatory agents is dependent on the time of exposure to and concentration of the drugs administered, desflurane at 6% could play a relevant role in organ protection during cardiac surgery. 

This study reveals that the nephroprotective effect of desflurane in short low-risk cardiac surgery is more intense than the one reported in previous studies. To achieve this effect, it is necessary to maintain expired gas concentrations more than twice the ones required for sevoflurane, according to the pharmacodynamics of each of these gases and their action on the brain, to achieve adequate hypnosis.

The potential clinical benefits of desflurane in aortic valve replacement surgery are two-fold, i.e., myocardial conditioning and nephropreservation. The latter would reduce the risk of infection. This would have a direct impact on mortality in this population with kidney failure through a nephroprotective effect that is independent from its cardioprotective effect on patients, as, in cardiac surgery, mortality increases with the occurrence of acute kidney failure.

### Limitations

One of the biases of our study was the non-randomization of patients due to the quasi-experimental design of the study; however, the anesthesiologists were permitted to select the type of hypnotic according to their usual practice. In addition, the statistical analysis showed no baseline differences between study groups. On the other hand, the same algorithm was used for the treatment of low cardiac output syndrome and for blood volume optimization in the two groups, thereby ensuring homogeneity. 

Another limitation of this study is that the mechanism by which halogenated anesthetics exerts nephroprotective effects, independent from its cardioprotective effects, is not identified.

Therefore, the potential association between the intraoperative use of desflurane and a lower elevation of biochemical parameters of renal dysfunction (NGAL and creatinine) needs further study with randomized studies that confirm our hypothesis.

## 5. Conclusions

The use of desflurane vs. propofol in aortic valve replacement surgery, including the extracorporeal circulation period, is associated with a decrease in creatinine and NGAL values. Further larger randomized studies are needed to confirm the potential causal relationship between anesthetics and renal function in aortic valve replacement surgery.

## Figures and Tables

**Figure 1 life-12-01172-f001:**
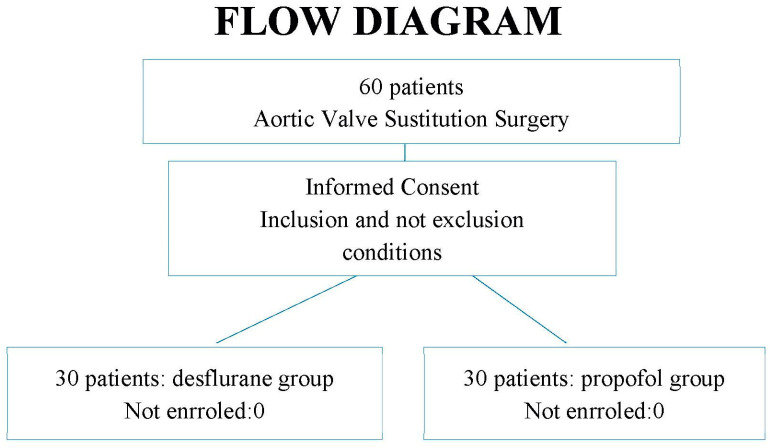
Flow diagram.

**Figure 2 life-12-01172-f002:**
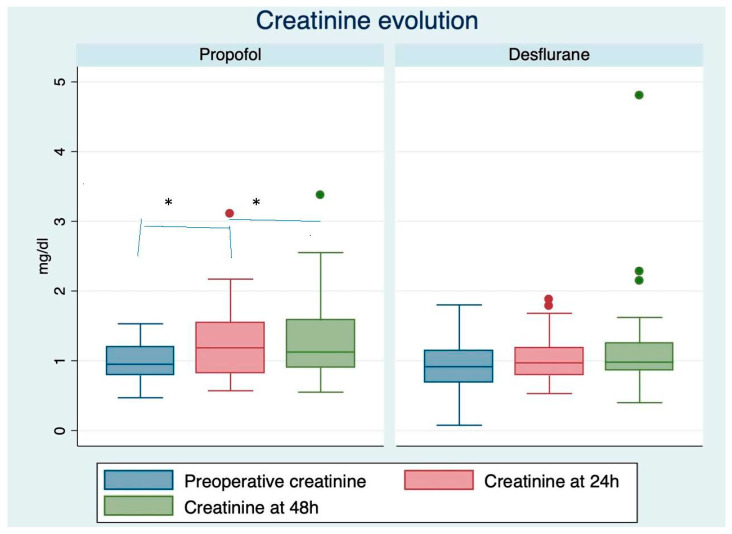
Variations in creatinine in relation to baseline values, at 24 and 48 h in each group (*t*-test). * *p* < 0.05.

**Figure 3 life-12-01172-f003:**
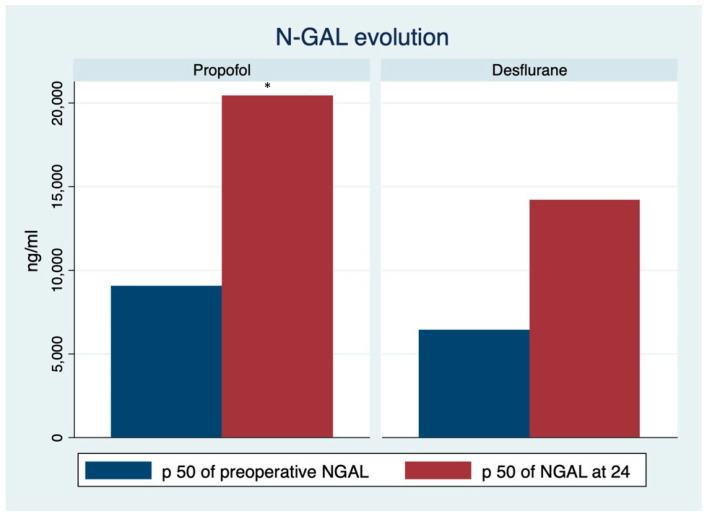
Variations in urinary NGAL in relation to baseline values and at 2 h in each group. Mann–Whitney–Wilcoxon test. * *p* < 0.05.

**Table 1 life-12-01172-t001:** Epidemiological variables. Ischemia time, ECC time, and age are represented by means, SD (standard deviation) and *t*-test *p*. Gender is compared using the z proportion test.

	Desflurane	Propofol	*p*
Clamping time (min)	43 +/− 8	45 +/− 12	0.84
ECC time (min)	83 +/− 5	82 +/− 7	0.95
Age (years)	65 +/− 4	68 +/− 6	0.33
Gender (M/F)	0.47	0.57	0.44

ECC: extracorporeal circulation.

**Table 2 life-12-01172-t002:** Hemodynamic and biochemical variables of myocardial damage and oxygen transport are represented by means and SD (standard deviation).

Variable	Propofol	Desflurane	*p*
CVP Pre	75 +/− 4	72 +/− 6	0.11
NT-ProBNP pre	846.3 +/− 345	1283.5 +/− 256	0.35/0.92
CPR pre	4.71 +/− 0.65	3.03 +/− 1.2	0.29/0.94
CKMb pre	13.11 +/− 2.1	1.95 +/− 1.3	0.12/0.08
CVS 24 h	69.1 +/− 2	71.5 +/− 1.5	0.29
NT-ProBNP 24 h	2338.9 +/−/− 154	2744.5 +/− 358	0.58/0.9
CRP 24 h	146.3 +/− 12.3	149.1 +/− 12.7	0.83
CKMMB 24 h	17.4 +/− 3.54	16.3 +/− 2.58	0.7/0.28
Tropo 24 h	2.8 +/− 0.32	7.8 +/− 0.33	0.25/0.52
Lactate 24 h	1.76 +/− 0.1	1.65 +/− 0.2	0.59/0.68
Norepinephrine 24 h	37%	38%	0.92
LCOS 24 h	30%	20%	0.37
MAP 24 h	74 +/− 12	77 +/− 9	0.35
HCT 24 h	33.7 + −/− 5	32.5 +/− 2.8	0.23
CVS 48 h	71 +/− 5	71 +/− 6	0.87/0.25
CKMMB 48	10.3 +/− 2.36	7.7 +/− 5.8	0.47/0.68
Tropo 48 h	1.73 +/− 0.45	2 +/− 0.38	0.42/0.87
Lactate 48 h	1.39 +/− 0.35	1.31 +/− 0.45	0.57/0.48
Norepinephrine 48 h	40%	20%	0.09
LCOS 48 h	23%	13%	0.32
MAP 48 h	77.7 +/− 6	79.6 +/− 8	0.47
HCT 48 h	31.3 +/− 1.2	31.3 +/−1.4	0.95

CVS: central venous saturation; CRP: C-reactive protein; Tropo: troponin I; LCOS: Low Cardiac Output Syndrome; MAP: mean arterial pressure; HCT: hematocrit; pre: preoperative.

**Table 3 life-12-01172-t003:** Variations in kidney stage between groups.

AKI 24	Propofol	Desflurane
0	17	23
Percentage in this stage	28.33	40
1	11	5
Percentage in this stage	18.33	8.33
2	2	2
Percentage in this stage	3.33	3.33
3	0	0
Percentage in this stage	0	0
**AKI 48**	**Propofol**	**Desflurane**
0	17	20
Percentage in this stage	28.33	33.33
1	12	9
Percentage in this stage	20.00	15.00
2	1	0
Percentage in this stage	1.67	0.00
3	0	1
Percentage in this stage	0.00	1.67

AKI: Acute Kidney Injury Scale.

## Data Availability

The data presented in this study are available on request from the corresponding author. The data are not publicly available due to ethical reasons.

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
