# Peer review of "Comparison of the Use of Desflurane vs. Propofol in Aortic Valve Replacement Surgery: Differences in Nephroprotection: An Explorative and Hypothesis-Generating Study"

_life, 2022, doi:10.3390/life12081172_

Round 1
Reviewer 1 Report
Guerrero et al present a study that describes the nephroprotective effect of desflurane in cardiac surgery.
Although the study has potential interest, I consider that significant changes have to be done to improve it:
-Introduction: clearly define primary and secondary objectives.
-Methods:
-Include number of approval of the committee
-Exclusion criteria: how do you define severe disease? Is criterium 4 included in number 5?
-What were the criteria to choose the anesthetic agent?
-Line 89: please reference the consensus
-Line 91: do you mean 15%?
-Line 107-109: Is it not clear at what time you collected biomarkers.
-What cardiac output monitoring did you use?
-Line 117: what classification did you employ for AKI? Please reference.
-Line 121: Prabhu et al. Please reference
-Statistical analysis is not described in methods section.
-Results: please describe how the variables are presented in tables (median?, mean?.). Use dipersion measures, as SD or IQR. Were there differences between groups in N-GAL and creatinine between groups? Is necessary to compare groups preoperatively, to check that were no differences and try to demonstrate differences in the postoperative period. Was not it the primary objective of the study?
-Discussion: What this study provides to new knowledge about this topic? What is the novelty? Please clarify
-Conclusions are not correct, only differences in biomarkers have been found (and comparisons in creatinine and N-GAL between groups have not been performed), not in the incidence of AKI. Please reformulate conclusions according to objectives.
Thank you very much for your work
Reviewer 2 Report
Please find attached the reviewer report form.

Reviewer 3 Report
Despite the actual topic and the considerable effort in study conduction and interpretation, it explicitly suffers from misconstruction and misconduction, leading to problematic and elusive conclusions. Few of the citations are incomplete, fragmented, or improper (E.g. 1.). The English wording of certain clinical categories may be misleading (i.e., repeated usage of kidney failure instead of kidney injury). The quasi-experimental study design (absence of randomization) is hardly acceptable, particularly in the setting of a limited study sample, and potentially introduces operator bias.
Having in mind the rapid dynamic of urine NGAL levels rise within the first 24h following CPB and its severalfold increase, the difference of 150 ng/ml in urine NGAL values used in the sample size calculation is unrealistically low, making the sample size calculation not only questionable but high likely incorrect, what undermines the whole study concept (2,3). Also, the authors neither used nor commented urinary NGAL values normalization according to urinary creatinine. The functional bias towards the clinical utility of urinary NGAL is indicative, as well (2,4).
Important data potentially influencing acute kidney injury following aortic valve replacement, such as the left ventricular ejection fraction, NYHA class, and percentage of insulin-requiring patients, are not presented. Important procedural data on the type of cardioplegia, type of aortic valve implanted, and the need for low energy cardioversion for rhythm restoration following CPB are not presented. That hinders any interpretation of cardio-specific enzymes behavior or relation with other parameters or outcomes.
I believe that the article in the presented form me be considered just as an explorative and hypothesis-generating study for a subsequent one encompassing a more significant number of participants applying combinations of the novel (including NGAL) with proper sample timing, as well as traditional markers of acute kidney injury.
References:
1. Prabhu A, Sujatha DI, Ninan B, Vijayalakshmi MA. Neutrophil gelatinase associated lipocalin as a biomarker for acute kidney injury in patients undergoing coronary artery bypass grafting with cardiopulmonary bypass. Ann Vasc Surg. 2010;24(4):525-531. doi:10.1016/j.avsg.2010.01.001
“About 30 patients undergoing CABG with CPB were prospectively studied. Blood was collected before bypass, at 4, 12, and 24 hr after CPB initiation, for the analysis of NGAL and oxidative stress markers.”
2. Bataille A, Tiepolo A, Robert T, et al. Reference change values of plasma and urine NGAL in cardiac surgery with cardiopulmonary bypass. Clin Biochem. 2017;50(18):1098-1103. doi:10.1016/j.clinbiochem.2017.09.019
“In patients who underwent coronary artery bypass grafting with normal post-operative kidney function, two-fold change in plasma NGAL and three to six-fold change in urine NGAL occur. In this specific clinical context, pathological variations must consider this biological "noise" for correct interpretation.”
3. Bennett M, Dent CL, Ma Q, Dastrala S, Grenier F, Workman R, Syed H, Ali S, Barasch J, Devarajan P. Urine NGAL predicts severity of acute kidney injury after cardiac surgery: a prospective study. Clin J Am Soc Nephrol. 2008 May;3(3):665-73. doi: 10.2215/CJN.04010907. Epub 2008 Mar 12. PMID: 18337554; PMCID: PMC2386703.
“in subjects who subsequently developed AKI, there was a robust 15-fold increase in urine NGAL at 2 h after CPB, which was even further accentuated at the 4 h (25-fold increase) and 6 h (26-fold increase) time points. In the AKI group, a statistically significant increase in urine NGAL from baseline was apparent up to 48 h after CPB. Similar findings were recorded when the urine NGAL measurements were corrected for urinary creatinine concentrations”
4. Ho J, Tangri N, Komenda P, et al. Urinary, Plasma, and Serum Biomarkers' Utility for Predicting Acute Kidney Injury Associated With Cardiac Surgery in Adults: A Meta-analysis. Am J Kidney Dis. 2015;66(6):993-1005. doi:10.1053/j.ajkd.2015.06.018
“Our meta-analysis of biomarkers in the early detection of AKI following cardiac surgery has 2 important findings. First, we found that current biomarkers have generally poor and at best moderate discrimination for AKI when measured within the first 24 hours after cardiac surgery in adults. Second, at present, there are comparatively few data for the discrimination of these biomarkers in the intraoperative period, a time of potential active management to mitigate kidney injury. Only u-NGAL has been studied more than once, but its intraoperative diagnostic performance was limited. Our findings highlight the need for further investigation into the early detection of cardiac surgery–associated AKI.”
Author Response
Please, see the attachment

Round 2
Reviewer 1 Report
Authors have clearly improved the manuscript after revision. I have notwithstanding minor recommendations:
-Page 55-61: I consider it is better to have this clarification in discussion, as a strength of the study .
-Figure 2: according to authors' explanation NGAL is measured at 2h, not after 24h.
-Conclusion: in this kind of study you can find associations, not causality. Please change
Thank you very much for your valuable work .
Reviewer 2 Report
Review report is attached.

Reviewer 3 Report
The authors promptly provided the revised article, implementing - to their best - corrections and updates according to reviewers' suggestions. Of particular importance is updating the article's title, indicating an explorative and hypothesis-generating study.
Considering this, I agree that the article merits publication in the corrected version.
Author Response
Thanks a lot for all your comments.